# Influence of three phases of El Niño-Southern Oscillation on daily precipitation regimes in China

Aifeng Lv[1,2], Bo Qu[1,2], Shaofeng Jia[1] and Wenbin Zhu[1]

[1] Key Laboratory of Water Cycle and Related Land Surface Processes, Institute of Geographic Sciences and Natural Resources Research, Chinese Academy of Sciences, Beijing 100101, China

[2] University of Chinese Academy of Sciences, Beijing 100049, China;

*Correspondence to*: Bo Qu (geo_qb@163.com) and Aifeng Lv (lvaf@163.com)

**Abstract**

In this study, the impacts of the El Niño-Southern Oscillation (ENSO) on daily precipitation regimes in China are examined using data from 713 meteorological stations from 1960 to 2013. We discuss the annual precipitation, frequency and intensity of rainfall events, and precipitation extremes for three phases (Eastern Pacific El Niño (EP), Central Pacific El Niño (CP), and La Niña (LN)) of ENSO events in both ENSO developing and ENSO decaying years. A Mann–Whitney U test was applied to assess the significance of precipitation anomalies due to ENSO. Results indicated that the three phases each had a different impact on daily precipitation in China and that the impacts in ENSO developing and decaying years were significantly different. EP phases caused less precipitation in developing years but more precipitation in decaying years; LN phases caused a reverse pattern. The precipitation anomalies during CP phases were significantly different than those during EP phases and a clear pattern was found in decaying years across China, with positive anomalies over northern China and negative anomalies over southern China. Further analysis revealed anomalies in frequency and intensity of rainfall accounted for these anomalies in annual precipitation; in EP developing years, negative anomalies in both frequency and intensity of rainfall events resulted in less annual precipitation while in CP decaying years, negative anomalies in either frequency or intensity typically resulted in reduced annual precipitation. ENSO events tended to trigger extreme precipitation events. In EP and CP decaying years and in LN developing years, the number of very wet days (R95p), the maximum rainfall in one day (Rx1d), and the number of consecutive wet days (CWD) all increased, suggesting an increased risk of flooding. On the other hand, more dry spells (DS) occurred in EP developing years, suggesting an increased likelihood of droughts during this phase. Possible mechanisms responsible for these rainfall anomalies are speculated by the summer monsoon and tropical cyclone anomalies in ENSO developing and decaying years.

**Key words: ENSO, daily precipitation, climate extremes, summer monsoon, tropical cyclone, China**

## 1 Introduction

The El Niño-Southern Oscillation (ENSO), a coupled ocean-atmosphere phenomenon in the tropical Pacific Ocean, exerts enormous influence on climate around the world (Zhou and Wu, 2010). Traditionally, ENSO events can be divided into a

warm phase (El Niño) and a cool phase (La Niña) based on sea surface temperature (SST) anomalies. An El Niño produces warming SSTs in the Central and Eastern Pacific, while La Niña produces an anomalous westward shift in warm SSTs (Gershunov and Barnett, 1998). Precipitation appears especially susceptible to ENSO events over a range of spatio-temporal scales and therefore has been the focus of many ENSO-related studies (Lü et al., 2011). Global annual rainfall over the land drops significantly during El Niño phases (Gong and Wang, 1999) and a wetter climate occurs in East Asia during El Niño winters due to a weaker than normal winter monsoon (Wang et al., 2008), but these anomalies are generally reversed during La Niña phases. Various studies also extensively document the teleconnections between ENSO and precipitation variation in China (Huang and Wu, 1989; Lin and Yu, 1993; Gong and Wang, 1999; Zhou and Wu, 2010; Lü et al., 2011; Zhang et al., 2013; Ouyang et al., 2014). Zhou and Wu (2010) found that El Niño phases induced anomalous strong southwesterly winds in winter along the southeast coast of China, contributing to an increase in rainfall over southern China. In the summer after an El Niño, less rainfall occurs over the Yangtze River, while excessive rainfall occurs in North China (Lin and Yu, 1993). During La Niña phases, annual precipitation anomalies are spatially opposite of those during El Niño phases in China (Ouyang et al., 2014). Typically, ENSO events progress over the previous winter and into the following spring/summer, thus influencing the climate of China or other areas in both the developing and decaying years (Ropelewski and Halpert, 1987; Lü et al., 2011). There was also a significant time lag in the responses of climate in China to ENSO evolution (Wu et.al., 2004). The delayed response of climate variability to ENSO provides valuable information for making regional climate predictions (Lü et al., 2011).

ENSO events are well-known for causing extreme hydrological events (Moss et al., 1994; Chiew and McMahon, 2002; Veldkamp et al., 2015) such as floods (Mosley, 2000; Räsänen and Kummu, 2013; Ward et al., 2014) and droughts (; Zhang et al., 2015) which in turn cause broad-ranging socio-economic and environmental impacts. Various approaches have been introduced to reveal these impacts at global and regional scales. For example, Ward et al. (2014) examined peak daily discharge in river basins across the world to identify flood-vulnerable areas sensitive to ENSO. Water storage is an index typically used to detect frequency and magnitude of droughts during ENSO events (Veldkamp et al., 2015; Zhang et al., 2015).

The physical mechanisms by which ENSO affects the climate of East Asia have also been discussed extensively in recent decades. Many studies have revealed that anomalous summer monsoons contribute to rainfall anomalies in East Asia during ENSO. A wet East Asian summer monsoon tends to occur after warm eastern or central equatorial Pacific SST anomalies during the previous winter (Chang et al., 2000). Floods and droughts during ENSO are also associated with the anomalous water vapor transport caused by the anomalous summer monsoon (Chang, 2004). On the other hand, tropical cyclones (TCs) over the western North Pacific (WNP) are also key contributors to rainfall events in China. When TCs move westward, a huge amount of moisture is transported into East Asia, accompanied by strong winds and heavy and continuous rainfall. By using

satellite-derived Tropical Rainfall Measuring Mission (TRMM) data, Guo et al. (2017) revealed that TCs occurring during the peak TC season (from July to October, JASO) contributed ~20% of monthly rainfall and ~55% of daily extreme rainfall over the East Asian coast. Strong TC activity suggests that there is excessive transport of water vapor into China.

Until recently, most studies have focused on changes in annual or seasonal total precipitation related to ENSO rather than changes in individual precipitation events. Changes in precipitation frequency and intensity are crucial for accurate assessment of ENSO impacts, but changes in mean precipitation cannot identify such changes. Recently, however, possible shifts in the characteristics of precipitation events (e.g. frequency and intensity) have been highlighted in studies of global climate change (Fowler and Hennessy, 1995; Karl et al., 1995; Gong and Wang, 2000). In China, it has been reported that the number of wet days per year decreased in recent decades even while total annual precipitation has changed very little (Zhai et al., 2005). Precipitation intensity has also changed significantly across China (Qu et al., 2016) and, as a result, drought and flood events occur more frequently (Zhang and Cong, 2014). Thus, separating out the impacts of ENSO events on precipitation frequency and intensity is critical to understanding ENSO-precipitation teleconnections in China. Although the link between hydrological extremes and ENSO is usually discussed in the context of the physical mechanisms that influence local precipitation (Zhang et al., 2015), direct precipitation indices such as the number of consecutive wet days and dry spells have rarely been addressed in these studies. Thus, our knowledge of how daily precipitation extremes respond to ENSO events is still very limited and requires a comprehensive set of precipitation indices that describe ENSO-induced precipitation extremes. A number of recent studies suggest that a new type of El Niño should be defined that is different from the canonical El Niño (Ren and Jin, 2011). This new discovered El Niño develops in regions of warming SSTs in the Pacific near the International Date Line (McPhaden et al., 2006) and has been called "Dateline El Niño" or "Central Pacific (CP) El Niño." Studies have revealed that CP El Niño appears to induce climate anomalies around the globe that are distinctly different than those produced by the canonical Eastern Pacific (EP) El Niño (Yeh et al., 2009). In addition, CP El Niño has been occurring more frequently in recent decades (Yu and Kim, 2013). Despite a long-term focus on ENSO-climate teleconnections, relatively little attention has been paid to the impacts of the new CP El Niño in China.

The current study aimed to provide a better understanding of how daily precipitation responds to the ENSO events over China. The main objectives of this work are to document (1) any changes in daily rainfall in China during three phases of ENSO events; (2) the number and duration of precipitation extremes occurring in ENSO developing and decaying years; (3) anomalous summer monsoon and TC activity induced by ENSO, and their relationships with rainfall anomalies. We discuss the total precipitation anomaly, anomalies of precipitation frequency and intensity patterns, and changes in precipitation extremes, and propose possible mechanisms responsible for the various rainfall anomalies.

## 2 Materials and methods

In this study, we used daily values of climate Chinese surface stations compiled by the National Meteorological Center in China. This dataset comprises detailed spatial coverage of precipitation across China, but only 400 stations were operational in the 1950s (Xu et al., 2011). Non-climatic noise can complicate the accuracy of the dataset analysis (Qu et al. 2016). Stations that experienced observation errors, missing values, or data homogeneity problems were omitted from analysis in this study, according to similar methods used by Qian and Lin (2005). Of the 819 meteorological stations across China, data from 713 were ultimately selected for analysis, which covered the time period of 1960–2013 (Fig. 1). Precipitation indices were calculated based on daily observations at the stations (Table 1). Annual total precipitation, as well as intensity and frequency of daily precipitation events, were used to formulate precipitation characteristics. Four other indices were introduced (Zhang et al., 2011), and used to analyse precipitation extremes in this study (Table 1). Precipitation indices were calculated for ENSO developing and decaying years. Indices for precipitation anomalies were analysed as follows:

$$A_{ij} = \frac{\overline{PI_{ij}} - \overline{PA_{ij}}}{\overline{PA_{ij}}},$$

(1)

Where $\overline{PI_{ij}}$ is the average of the $i$ precipitation index at the $j$ meteorological station during a specific time period, and $\overline{PA_{ij}}$ is the average of the $i$ precipitation index at the $j$ station for a multi-year average (1971–2000).

To quantify the variability of the summer monsoon and ENSO impacts over China, the monsoon index proposed by Wang and Fan (1999) was used in this study. It is defined as the 850hPa wind speed averaged over 5 ºN-15 ºN, 100-130 ºE minus the wind speed averaged over 20 ºN-30 ºN, 110-140 ºE, and is frequently used to study interannual and decadal variability of summer monsoons over the western North Pacific-East Asian region (WNP-EA). For TC activity, the best-track dataset from the Joint Typhoon Warning Center was obtained at http://www.metoc.navy.mil/jtwc/jtwc.html. Following the methods by Kim et al. (2011), TC genesis and track density was generated for ENSO developing and decaying years for the period 1960-2013. Track density anomalies are defined as annual average TC frequency during a specific type of ENSO event minus the long-term mean value in each 2°×2° grid box.

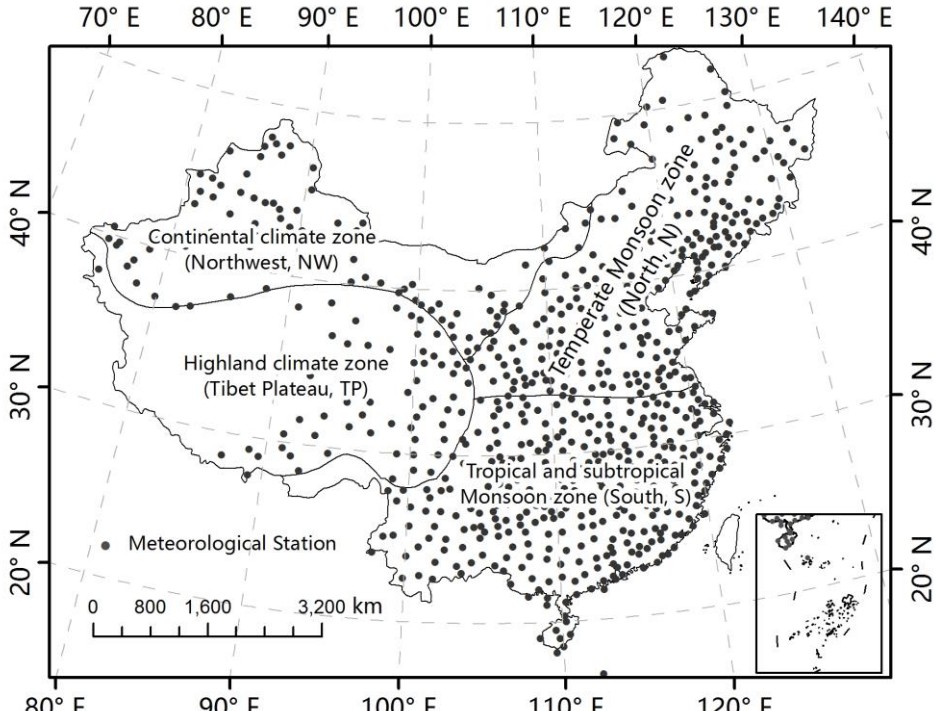

**Fig. 1.** Distribution of the 713 meteorological stations used in this study. China is divided into four regions based on climatic
type. The two non-monsoon regions are the continental climate zone (Northwest, NW) and the highland climate zone (Tibet
Plateau, TP). The monsoon region is divided into two regions: the tropical and subtropical monsoon zone (South, S) and the
temperate monsoon zone (North, N).
**Table 1.** Definitions of precipitation indices used in this study

| Index | Descriptive name | Definition | Unit |
|---|---|---|---|
| P | Annual precipitation | Annual total precipitation | mm |
| Intensity | Daily intensity index | Average precipitation per rain event (day with precipitation > 0) | mm/day |
| Frequency | Number of rainy days | Annual number of rainy days | day |
| Rx1d | Maximum 1-day precipitation | Annual maximum 1-day precipitation | mm |
| R95p | Very wet day precipitation | Annual total precipitation when precipitation greater than 95th percentile of multi-year daily precipitation* | mm |
| DS | dry spells | Number of consecutive dry days no less than 10 | count |
| CWD | consecutive wet days | Number of consecutive rainy days no less than 3 | count |

*95th percentile of multi-year daily precipitation is the 95th quantile of the daily precipitation distribution over 1971–2000
(Percentiles near 100 represent extremely intense precipitation).
In this study, two indices, created by Ren and Jin (2011) by transforming the traditionally-used Niño3 and Niño4 indices,
were used to distinguish between CP and EP El Niño phases. La Niña years were identified using the methods of McPhaden
and Zhang (2009). The ENSO events (1960–2013) analyzed in this study are displayed in Table 2. As an EP El Niño evolves,

positive SST anomalies expand latitudinally and negative signals expand eastward, reaching a maximum amplitude in autumn and winter (Feng et al., 2011). The first year was defined as the developing year of an EP El Niño in this study. Warm SST anomalies disappear and are replaced by cool anomalies in the eastern Pacific during summer of the next year (defined as the decaying year). Similarly, the emerging and vanishing years of CP El Niño and La Niña events are defined as the developing and decaying years, respectively.

**Table 2.** ENSO emerging years from 1960 to 2013

| Phase | Eastern Pacific (EP) El Niño | Central Pacific (CP) El Niño | La Niña (LN) |
|---|---|---|---|
| Year | 1963 1965 1969 1972 1976 1982 1986 1991 1997 2006 | 1968 1977 1987 1994 2002 2004 2009 | 1964 1967 1970 1973 1975 1984 1988 1995 1998 2007 2010 |

The significance of ENSO-induced precipitation anomalies is tested using a Mann–Whitney U approach. The Mann–Whitney U test is a nonparametric test applied to site data which does not conform to normality even after several transformations are performed (Teegavarapu et al., 2013). It tests whether two series are independent from each other. One series represents precipitation during an ENSO event phase (EP, CW, or LN), and the other series represents precipitation during average years. This test was applied to evaluate the significance of precipitation anomalies at a significance level of 5%.

**3 Results**

**3.1 Annual rainfall anomalies**

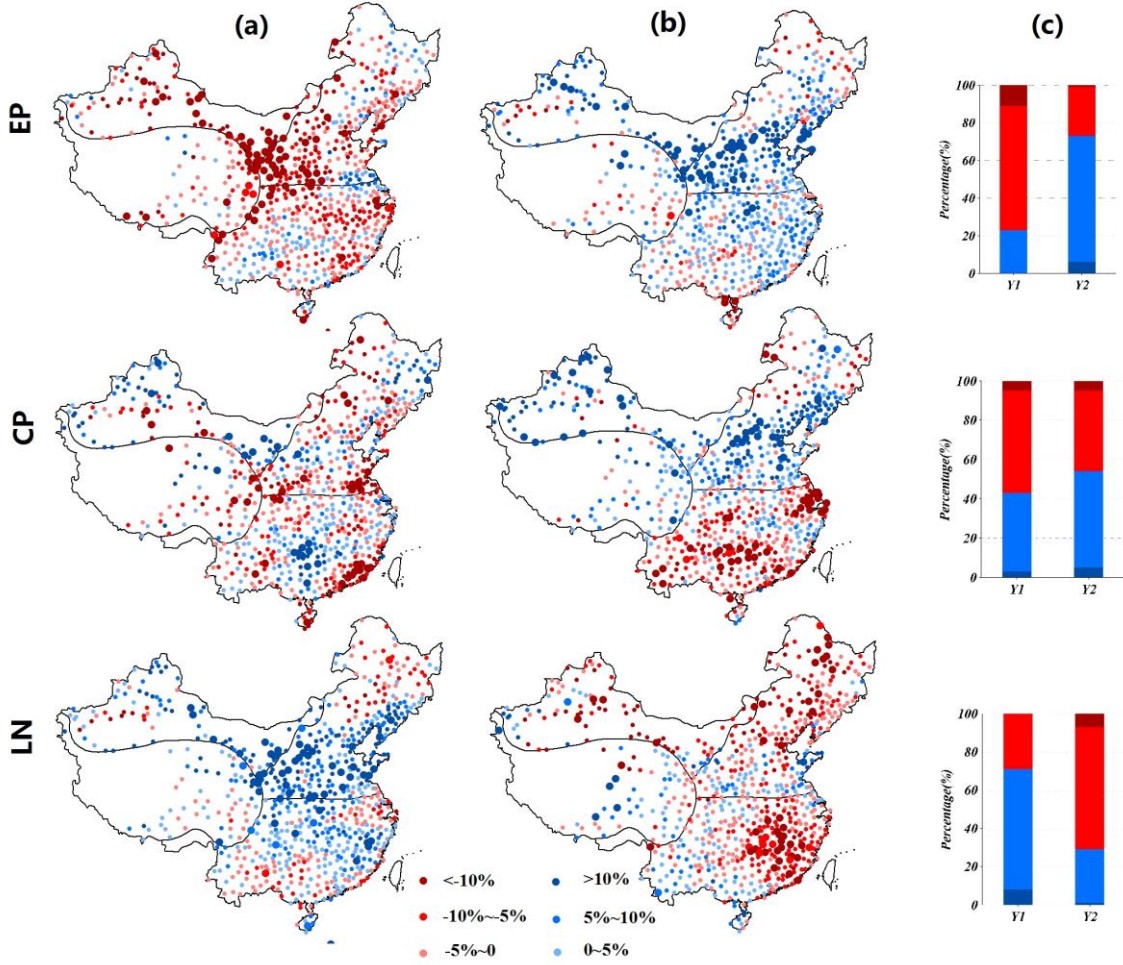

138

**Fig. 2** Anomalies of annual precipitation in developing years (a) and decaying years (b) of EP, CP, and LN phases. Stations experiencing significant anomalies are represented by large points. The percentage of stations experiencing increases or decreases in the number of rainfall anomalies are shown in (c), with significant increase (blue), increase (light blue), decrease (light red), and significant decrease (red). Y1, Y2 represent developing years and decaying years, respectively.

In EP developing years, 628 stations across China (~80%) had negative anomalies, and 80 of these stations the anomalies were significant. These significant stations were mainly located in the continental climate zone (NW) and the temperate monsoon zone (N) (Fig. 2). All sub-regions experienced negative average annual precipitation anomalies (Fig. 3), especially in the NW region where precipitation was 12.83% lower than the mean. Large positive anomalies of annual precipitation were found during LN developing years (Fig. 2); more than 70% of the stations showed positive anomalies, of which 10% were significant. Similarly, the stations with significant anomalies were mainly in the NW and N regions. In CP developing years, precipitation anomalies were quite different from those in EP developing years (Fig. 2). The proportion of stations with negative anomalies was 57%, but with no clear pattern of distribution.

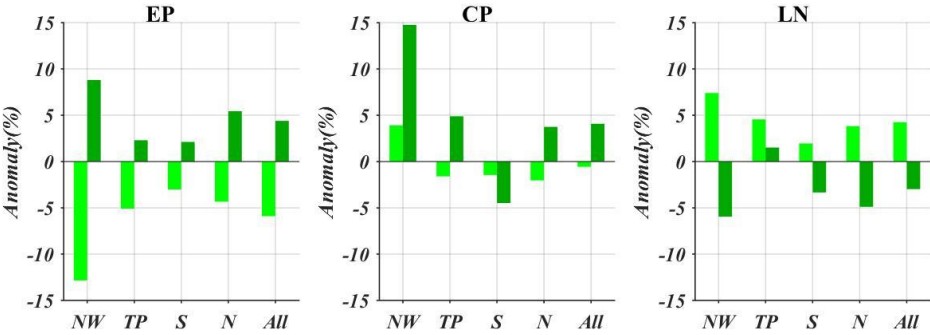


**Fig. 3** Average annual precipitation anomaly by sub-region during EP, CP, and LN phases. Light color represents developing
years and dark color represents decaying years.
The impacts of EP phases on precipitation in decaying and developing years displayed opposite patterns. Positive anomalies
were detected across China during decaying years (Fig. 2), especially in the NW and N regions at 8.8% and 8.9% higher than
the mean, respectively (Fig. 3). And negative anomalies were common across China in LN decaying years (Fig. 2). In the
NW region, average annual precipitation was 5.95% lower than the mean. These results suggested in both the decaying years
of EP and the developing years of LN, more water vapor would be transported from the Pacific Ocean to China, while in the
decaying years of LN and the developing years of EP, drier conditions would prevail. In the CP decaying phases, average
annual precipitation in the NW, Tibet Plateau (TP), and N regions was much greater than the mean, but lower than the mean
in the subtropical monsoon zone (S) (Fig. 3).
**3.2 Rainfall frequency and intensity anomalies**

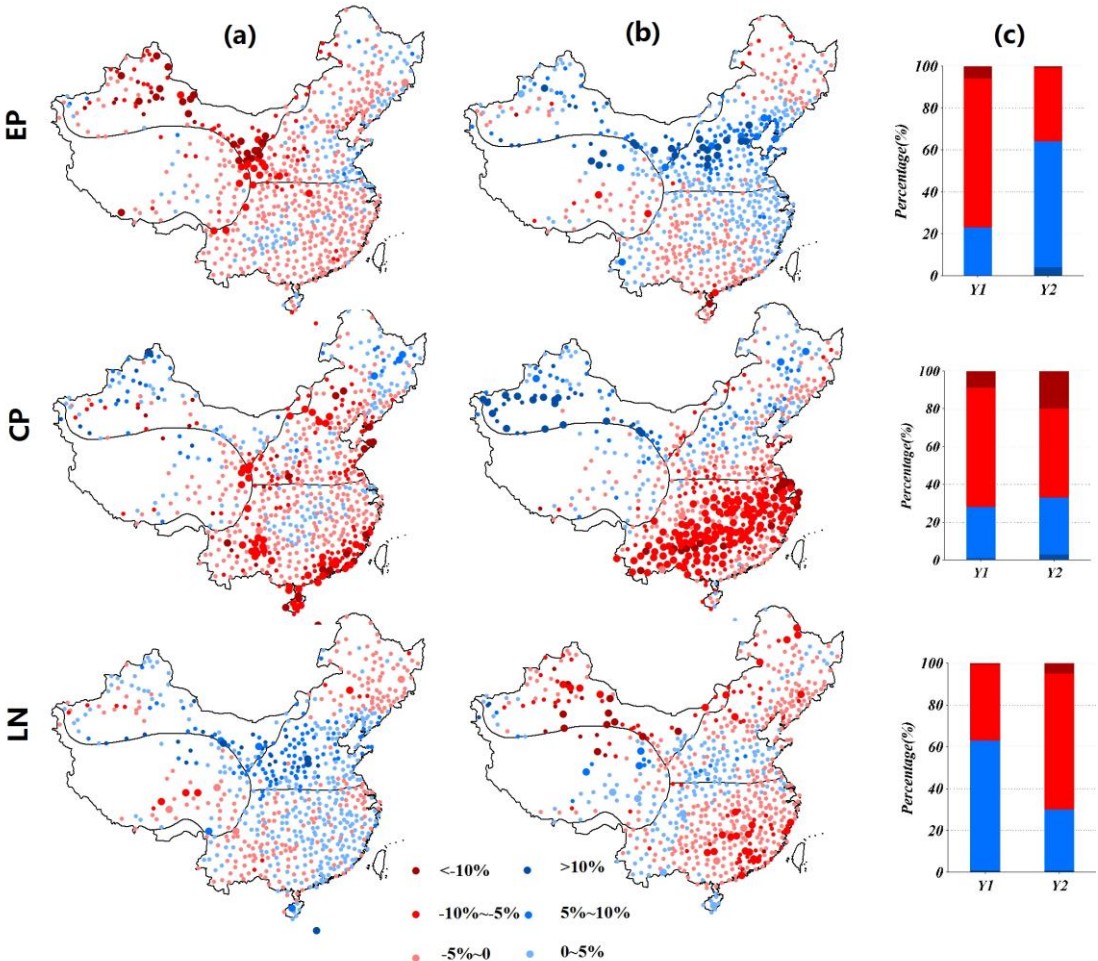


**Fig. 4** Anomalies of precipitation frequency in developing years (a) and decaying years (b) of EP, CP, and LN phases. Stations
experiencing significant anomalies are represented by large points. The percentage of stations experiencing anomalies of
precipitation frequency are shown in (c), with significant increase (blue), increase (light blue), decrease (light red), and
significant decrease (red). Y1, Y2 represent developing years and decaying years, respectively.


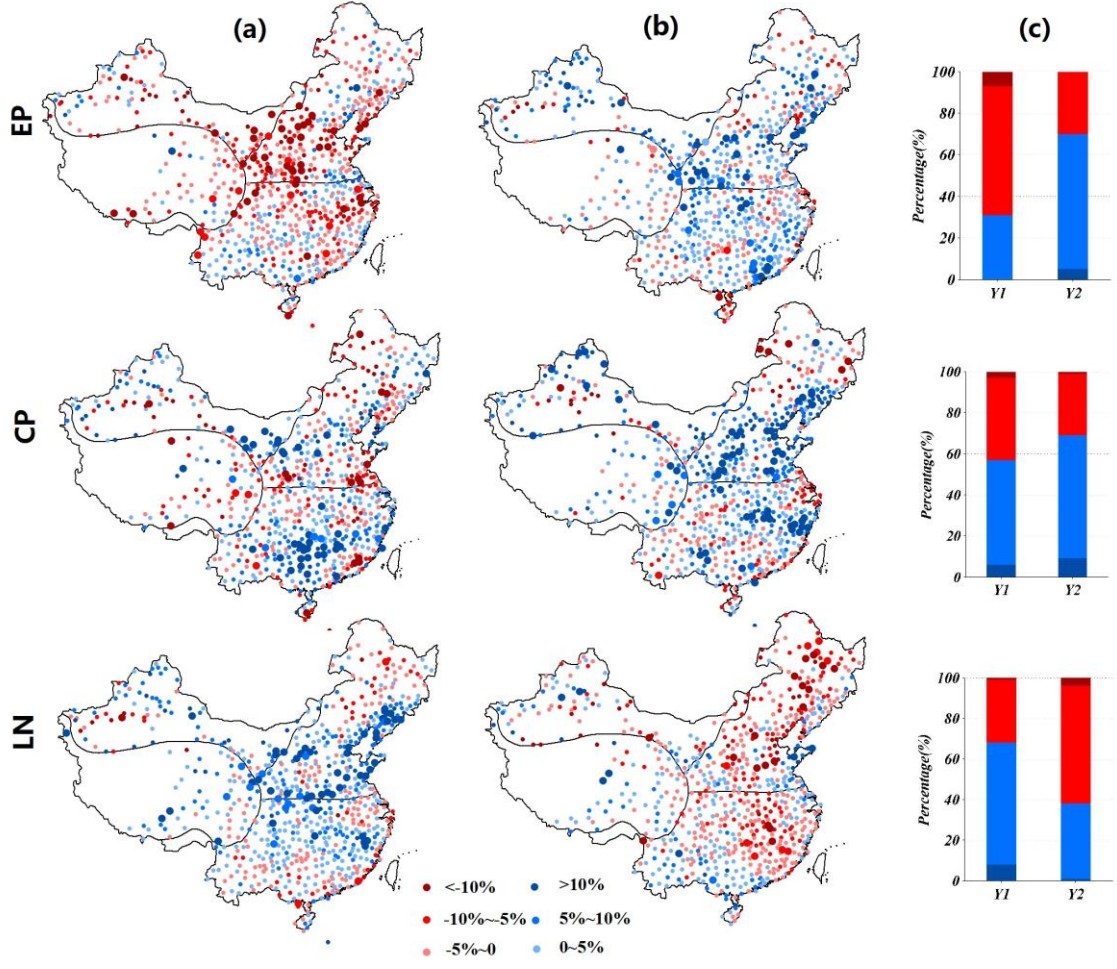


**Fig. 5** Anomalies of precipitation intensity in developing years (a) and decaying years (b) of EP, CP, and LN phases. Stations experiencing significant anomalies are represented by large points. The percentage of stations experiencing anomalies of precipitation frequency are shown in (c), with significant increase (blue), increase (light blue), decrease (light red), and significant decrease (red). Y1, Y2 represent developing years and decaying years, respectively.

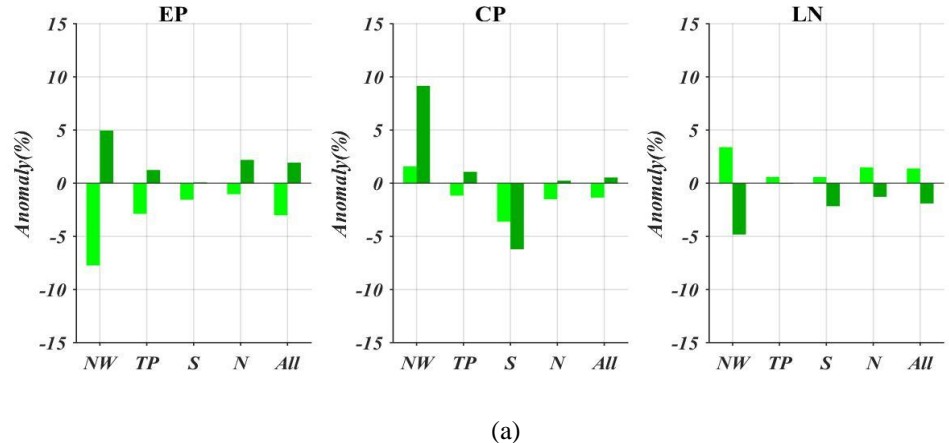

176

177                                                    (a)

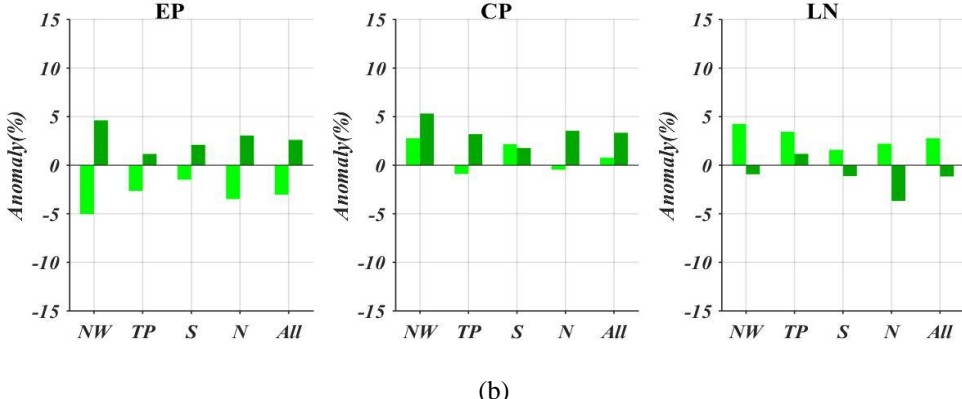

(b)

**Fig. 6** Average anomalies of precipitation frequency (a) and precipitation intensity (b) by sub-region during EP, CP, and LN phases. Light color represents developing years and dark color represents decaying years.

In EP developing years, only negative anomalies of precipitation intensity and frequency occurred, with decreases of 3.04% and 3.01%, respectively, across all of China (Fig. 6). Stations with significant decreases in precipitation frequency were mainly located in the NW region (Fig. 4) and stations with significant decreases in precipitation intensity were mainly located in the N region (Fig. 5). In contrast, anomalies of precipitation intensity and frequency in EP decaying years were positive, presenting a reverse pattern to the developing phase. Anomalies of precipitation intensity and frequency were also positive in LN developing years, with stations of significance concentrated in the N region.

In the CP phases, anomalies of precipitation intensity and frequency displayed more complex patterns than those in the EP years. In developing years, slightly more than half of the stations experienced positive anomalies of precipitation intensity (Fig. 5), while more than 70% experienced negative anomalies in precipitation frequency (Fig. 4). Of the stations experiencing negative precipitation frequency anomalies, 64 were significant (Fig. 4) and were concentrated in the S and N regions (Fig. 4). Precipitation frequency anomalies also formed a clear distribution pattern in CP decaying years (Fig. 4). Of all the meteorological stations, 145 (20%) experienced significant negative anomalies and were concentrated in the S region. In contrast, all regions experienced positive anomalies of precipitation intensity.

In general, anomalies of total precipitation tend to result from changes in both the frequency and intensity of precipitation events. Combined with the analysis in section 3.1, the results suggest that increases in precipitation frequency and intensity during EP decaying years and LN developing years resulted in the positive anomalies of annual precipitation across China during these phases. And the decreases in precipitation frequency and intensity during EP developing years and LN decaying years resulted in the negative anomalies of annual precipitation. But, in the CP phases, few regions displayed such clear relationships between anomalies in total precipitation and precipitation events. For example, in the N region, precipitation frequency changed very little, and the observed positive anomalies of annual rainfall in CP decaying phases appear to have resulted from increased precipitation intensity. Likewise, in the S region, precipitation intensity increased by 1.77% even though the precipitation frequency and total precipitation decreased.

 **3.4 Precipitation extremes**

ENSO can trigger extreme hydro-climatological events such as floods, droughts, and cyclones (Zhang et al., 2013). Table 3
shows the average percent change in the number of extreme precipitation events (anomalies of precipitation extremes) in
sub-regions and the whole of China, based on data from all meteorological stations.

208         **Table 3.** Average anomalies of precipitation extremes during EP, CP, and LN phases (%).

| Years | Phases | Index | NW | TP | S | N | All |
|---|---|---|---|---|---|---|---|
| Developing years | EP | Rx1d | -7.54 | -2.23 | -0.34 | -0.32 | -2.29 |
| | | R95p | -20.68 | -7.02 | -4.76 | -5.55 | -8.78 |
| | | DS | 3.38 | 3.93 | 1.94 | 1.59 | 2.67 |
| | | CWD | -15.42 | -6.07 | -0.95 | -3.70 | -5.96 |
| | CP | Rx1d | 4.89 | -1.00 | 1.02 | -2.73 | 0.27 |
| | | R95p | 5.79 | -0.99 | 0.57 | -2.93 | 0.28 |
| | | DS | -2.12 | 0.83 | -0.10 | 2.18 | 0.34 |
| | | CWD | 9.91 | -2.11 | -3.66 | -3.33 | -0.42 |
| | LN | Rx1d | 4.87 | 4.73 | 3.40 | 2.84 | 3.90 |
| | | R95p | 10.79 | 8.90 | 4.76 | 8.10 | 7.97 |
| | | DS | -1.21 | 3.58 | -0.25 | 0.26 | 0.71 |
| | | CWD | 8.59 | 2.82 | 0.99 | 4.31 | 3.89 |
| Decaying years | EP | Rx1d | 9.52 | 1.30 | 2.43 | 1.94 | 3.43 |
| | | R95p | 15.26 | 5.27 | 4.22 | 7.24 | 7.53 |
| | | DS | -3.83 | 1.70 | 0.26 | 1.76 | 0.21 |
| | | CWD | 5.96 | 2.22 | -1.48 | 6.72 | 3.18 |
| | CP | Rx1d | 7.54 | 4.36 | 2.39 | 0.28 | 3.39 |
| | | R95p | 23.32 | 7.99 | -0.33 | 7.13 | 8.64 |
| | | DS | 4.08 | -3.00 | 13.24 | -1.88 | 3.05 |
| | | CWD | 17.78 | 4.14 | -4.78 | 1.11 | 3.71 |
| | LN | Rx1d | -2.50 | 2.22 | -1.00 | -4.17 | -1.29 |
| | | R95p | -4.73 | 2.14 | -3.31 | -9.06 | -3.67 |
| | | DS | 1.85 | -2.48 | 1.35 | 0.87 | 0.30 |
| | | CWD | -7.86 | -0.70 | -1.81 | -3.04 | -3.06 |

During EP developing years and LN decaying years China experienced markedly negative anomalies in very wet daily
rainfall, as expressed by the R95p index, and positive anomalies during EP decaying years and LN developing years. These
impacts of the EP and LN phases on R95p were observed in nearly all sub-regions of China. An R95p positive anomaly was
also observed in CP decaying years, but only in the NW, TP, and N regions. In CP developing years, the R95p identified no
significant anomalies. The Rx1d index, a measure of maximum daily rainfall, revealed similar patterns to those identified by
the R95p index. Positive R95p and Rx1d values during EP and CP decaying years and LN developing years indicate an
increased likelihood of extreme precipitation events during these years than normal.
As shown in Table 3, negative anomalies of consecutive wet days (CWD) occurred in EP developing years and LN decaying
years across China while the opposite pattern occurred in EP and CP decaying years and in LN developing years. The CWD
is a measure of wet conditions that is closely related to soil moisture and river runoff. A greater number of CWDs will
enhance soil moisture, runoff, and the risk of floods. The NW region, a continental climate zone, is the most sensitive of
China's sub-regions to ENSO events in terms of CWDs. In EP developing years, the N, TP, and NW regions experienced
large decreases in CWDs (5.69%, 6.07%, and 15.42%, respectively). Such decreases have the potential to induce droughts in
these sub-regions as soil moisture decreases. But the dry conditions in these sub-regions reversed in EP decaying years.
Although a positive anomaly occurred in annual precipitation during CP decaying years in the N region, it experienced
smaller CWD anomalies. This was possibly due to the increase in intensity of rainfall events.
Dry spells (DS) are extended periods of 10 days or more of no precipitation and are a strong predictor of droughts. As shown
in Table 3, all sub-regions of China experienced positive anomalies in DS during EP developing years, displaying an inverse
pattern to that observed for CWDs discussed above. In other words, fewer CWDs and more DS occurred simultaneously and
indicated an increased risk of drought. Negative anomalies in DS were observed in the NW and N regions during EP
decaying years. In CP decaying years, DS displayed dipole anomalies across China which were opposite of observed CWD
patterns during the same period. But during the same years, DS anomalies were positive in the NW region even though
annual precipitation had increased. DS displayed far weaker anomalies during both LN developing years and decaying years.
**4 Discussion**
Summer monsoons over East Asia (EA) consist of staged progressions of zonally-oriented rain belts as fronts advance and
retreat. Huang and Wu's (1989) study first revealed that these summer monsoon rain belts are closely linked with ENSO
cycle phases. Figure 7 shows the mean WNP-EA monsoon index and its significant difference from average conditions

236 (1971-2000).

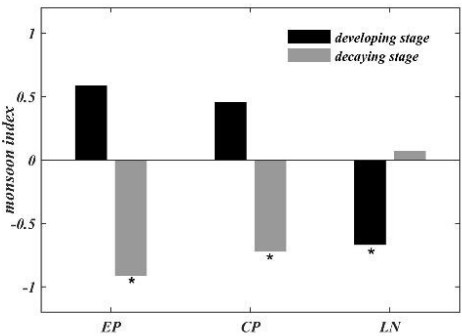


**Fig. 7** The western North Pacific-East Asian (WNP-EA) summer monsoon index during different ENSO phases (average for
1971-2000 is -0.007). * 95% significance
Results reveal that the WNP-EA monsoons tend to be weak during EP decaying years (fig. 7). Wang et al. (2001) showed
that a weak WNP-EA monsoon usually features enhanced rainfall along the monsoon's front over East Asia. As for the
mechanism, anomalous anticyclones in the subtropical WNP are the key systems linking the ENSO and the East Asian
climate (Feng et al., 2011; Wang and Chan, 2002; Wang et al., 2001; Yuan et al., 2012). An anomalous WNP anticyclone
during a weak WNP-EA monsoon brings plentiful moisture to southern China; meanwhile, it can also shift the ridge of the
sub-tropical high westward (Feng et al., 2011). When summer monsoons advance and retreat along WNP anticyclone fronts,
heavy and continuous rainfall typically develop along the monsoon fronts (Chang, 2004). In this study, our examination of
variations in precipitation anomalies reveals that rainfall is largely enhanced over the NW and N region during EP decaying
years (fig. 2).
Weak WNP-EA summer monsoons also tend to occur during CP decaying and LN developing years (fig. 7). Typically,
anomalous WNP anticyclones originate and develop during the previous autumn in the El Niño developing year and persist
until the following spring and summer before intensities decrease (Wang et al., 2003). Yuan et. al (2012) found that WNP
anticyclones display distinct location, intensity, and lifetime evolutions in the CP and EP El Niños due to the different
anomalous SSTs in the equatorial Pacific. The EP El Niño tends to create stronger, wider, and longer-lived WNP
anticyclones than the CP El Niño (Shi and Qian, 2018). In terms of rainfall pattern, the CP El Niño induced asymmetric
anomalies that do not follow the patterns seen in the EP El Niño (fig. 2). For example, during CP decaying years, the S
region experienced a negative annual precipitation anomaly. Anomalous WNP anti-cyclones may explain this incongruity
between the influences of the EP and CP El Niño phases, reflecting the potential for changes in atmospheric diabatic forcing
over the tropics. In contrast, weak WNP-EA summer monsoons during LN developing years possibly correlate with the
disappearance of EP during decaying years when WNP anticyclones tend to re-invigorate and extend northwestward and
inland (Feng et al., 2011). Precipitation anomalies in China also reveal a marked consistency between EP decaying years and
LN developing years (fig. 2).
Fig. 7 further reveals that strong WNP-EA summer monsoons occur during EP or CP developing years, although not
significantly. However, only the EP developing stage induces a negative rainfall anomaly over China (fig.2). Similar results
were observed by Wu et. al (2003), who documented seasonal rainfall anomalies in East Asia, finding that the rainfall
correlation distribution displayed pronounced differences between developing and decaying ENSO years. A reverse
monsoon signal between developing and decaying years suggests that WNP anticyclones respond in terms of location and
intensity to the evolution of SST anomalies over the tropical Pacific (Chang, 2004).
Using a climate model, Chou et al. (2012) found that changes in precipitation frequency and intensity are closely associated
with changes in atmospheric water vapor and vertical motion. As demonstrated by Chou et al. (2012), an increase in water
vapor reduces the magnitude of the vertical motion required to generate the same strength of precipitation, resulting in an
increase in precipitation frequency and intensity. Therefore, large amounts of water vapor transported during EP and CP
decaying years, or during LN developing years, when WNP-EA summer monsoons are relatively weak may enhance both
precipitation frequency and intensity. On the other hand, atmospheric vertical motion also tends to be intense during these
periods, as summer monsoons over China feature strong southerly winds (Chen et al., 2013). This leads to further anomalous
R95p, Rx1d, and CWD, resulting in increased flood risk during these years (table 3). However, a reduction in water vapor
availability and vertical motion may occur during EP and CP developing years as WNP-EA summer monsoons tend to be
strong (fig. 7), resulting in a negative anomaly in frequency and intensity of precipitation (fig. 4 & 5).
In addition, the relative stability of the atmosphere tends to reduce the frequency and intensity of precipitation by reducing
vertical motion (Chou et al., 2012). The WNP subtropical high is a prime circulation system over the WNP-EA and
anomalies of location and intensity largely affect summer monsoon activities in East Asia (Wang et al., 2013). Huang and
Wu (1989) found that when the location of a subtropical high is shifted unusually northward, hot and dry weather occurs in
East China due to the dominance of the stable atmosphere. The location and intensity of subtropical highs are also closely
associated with the development of WNP anticyclones, and the northward shift usually coincides with strong WNP-EA
summer monsoons (Wang et al., 2001). Therefore, anomalous WNP subtropical highs possibly exacerbate negative
precipitation frequency anomalies and positive DS anomalies in the S region during LN decaying years (fig4 & table 3). This
may also explain the strong reductions in precipitation frequency in the S region during CP decaying years (fig.4), because
WNP anticyclones display different anomalies than EP phases.
ENSO is one of the most important factors affecting TC activity over the WNP (Wu et. al, 2012). In this study, the
modulation of TC activity by ENSO was analyzed during developing and decaying phases for the period 1960-2013. Fig. 8
shows ENSO-induced anomalies in terms of TC number and location of formation. Track density anomalies are shown in fig.

291  9.

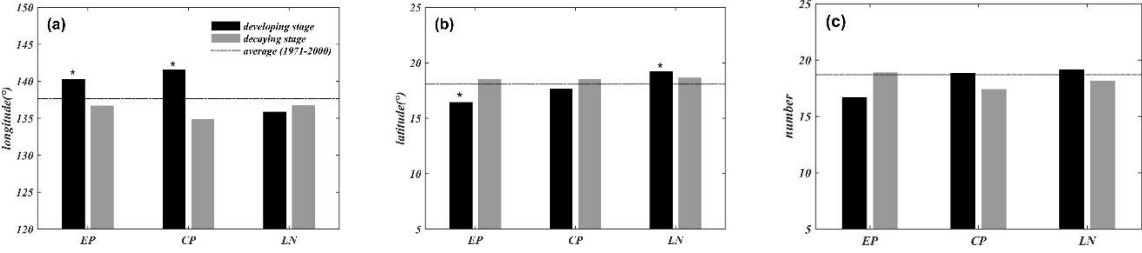

**Fig. 8** Anomalies in TC genesis longitude (a), latitude (b), and number (c) during JASO over the WNP. The dashed line
indicates the average value for 1971-2000. * 95% significance.

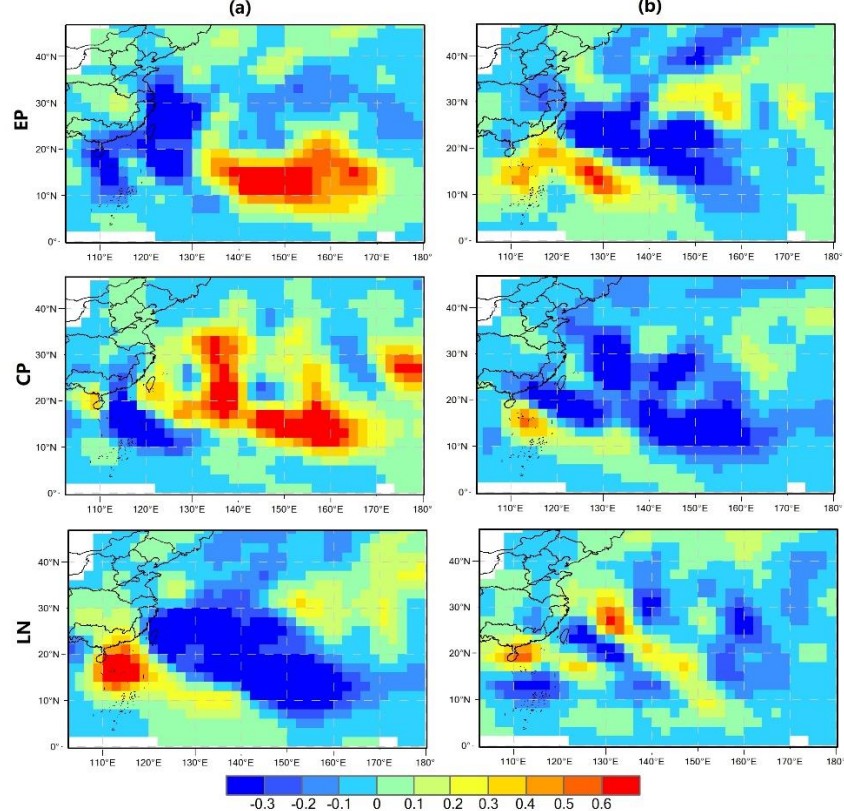

**Fig. 9** Track density anomalies during JASO in developing years (a) and decaying years (b) of EP, CP, and LN phases.

Although the total number of TSs formed in the WNP did not vary significantly from year to year, TCs tended to form further east and south during EP developing years (fig. 8). This shift in the location of TC genesis constrained TCs westward propagation into East Asia (Wang and Chan, 2002). Therefore, track density was largely reduced over the coast of China (fig. 9). This suggests that the EP developing stage induces a smaller impact by TC on rainfall in China. Conversely, during LN developing years, TCs tend to form further north (fig. 8), and the track density shows a remarkable increase in the South China Sea (fig. 8). The enhancement of TC activity tends to induce heavier rainfall events, leading to positive anomalies of precipitation intensity, Rx1d, R95p, and CWD during such years (fig. 4 & 5, table 3). TCs during CP developing years also form at higher latitudes than during EP phases, but the average latitude is lower than during the LN phase (fig. 8). In contrast, the CP developing stage increases track density from the central–western Pacific to the eastern China coast and decreases it over the South China Sea (fig. 9). Kim et al. (2011) revealed that a shift in TC genesis location during CP years is closely associated with anomalous westerly winds induced by the westward shift in ocean heating, and that this shift further provides more favorable conditions for westward TC propagation. Zhang et. al (2012) also claimed that TCs during CP summers are more likely to make landfall over East Asia because of a westward shift in subtropical highs and a northward shift in TC genesis. However, TCs during CP developing years do not exert significant rainfall anomalies over China. The anomalous WNP-EA summer monsoons induced by ENSO may further explain this discrepancy. As discussed above, the strong WNP-EA monsoons during CP developing years do not induce negative rainfall anomalies over China (fig. 7). This suggests

that enhanced TC activity may cause a reduction in rainfall along monsoon fronts, resulting in neutral conditions over China.
However, further studies are needed to examine how the CP developing stage influences rainfall over China.
In contrast, no significant shifts in the locations of TC genesis occurred during decaying years (fig. 8). This suggests that the
impact of ENSO on TC formation may decrease after ENSO maturation. However, nearly opposite TC track density patterns
occur over the WNP during developing and decaying years of an EP or CP (fig.9). For example, in EP decaying years, TC
activity increased in the South China Sea and decreased from the western Pacific to the eastern China coast. This shift in
track density affected water vapor transport and contributed to a reversed pattern of rainfall anomalies between developing
and decaying years.
**5 Summary**
Using a nonparametric hypothesis test, this study investigated the impacts of three different ENSO phases on daily rainfall
regimes in China during the past half century. Rainfall data collected from meteorological stations across the country
revealed that the impacts of the three phases were significantly different from each other on a daily time scale. ENSO events
triggered large changes in the frequency and intensity of precipitation events and in the occurrence of precipitation extremes.
This finding is significant because past studies examining teleconnections between ENSO events and climate variation in
China have primarily focused on annual and/or monthly rainfall rather than on daily precipitation events. Since ENSO events
can be predicted one to two years in advance using various coupled ocean/atmosphere models (Lü et al., 2011), this study
can provide a means of climate prediction on a daily time scale and enable the prioritization of adaptation efforts ahead of
extreme events.
Previous studies have revealed that some regions in China are especially vulnerable to ENSO events via teleconnections,
such as the South China Sea (Qu et al., 2004; Rong et al., 2007; Zhou and Chan, 2007; Liu et al., 2011) and the Yangzi River
(Huang and Wu, 1989; Tong et al., 2006; Zhang et al., 2007; Zhang et al., 2015). However, using daily precipitation indices,
we found that the continental climate zone (NW) is more sensitive than other regions to ENSO events due to the its high
incidence and magnitude of anomalous precipitation events (Fig. 4 & 5 and Table 3). For example, the NW region
experienced the largest R95p and CWD anomalies during all ENSO event phases. In an earlier study on daily river
discharges at a global scale, Ward et al. (2014) found that ENSO has a greater impact on annual floods in arid regions than in
non-arid regions. In China, Hui et al.(2006) analyzed interdecadal variations in summer rainfall in repsonse to the SST
anomaly over the Niño-3 region. They found that summer rainfall in northwestern China was well-predicted by ENSO
events between 1951–1974 (Hui et al., 2006). But little research has been conducted on the mechanisms behind climatic
responses to ENSO events in China's continental climate zone because most studies have focused on monsoon zones
(Matsumoto and Takahashi, 1999; Wen et al., 2000; Wang et al., 2008; Zhou and Wu, 2010).
Although the primary physical processes and mechanisms responsible for precipitation anomalies have been discussed in the
context of summer monsoons and TC activity, approaches to understanding the forces influencing daily precipitation events
coinciding with ENSO are more complex than those directed toward precipitation influences on a monthly or annual scale.
This complexity can be illustrated by the observation that in CP decaying years, the N region experienced a positive anomaly
of annual precipitation due to an increase in precipitation intensity, but the S region experienced a negative anomaly due to a
large decrease in precipitation frequency. Therefore, even though some physical mechanisms may explain precipitation
variabilities related to ENSO events, there is a need for more research on the mechanisms driving atmospheric circulation to
advance our understanding of these influences over temporal and spatial scales. In addition, the year-to-year variability of
East Asian summer monsoons are likely influenced by complex air–sea–land and tropical–extratropical interactions in
addition to ENSO events. These interactions may include Tibetan Plateau heating, Eurasian snow cover, and polar ice
coverage (Wang et al., 2000). Other factors that may contribute to precipitation anomalies in China during ENSO events
include forces that generate large-scale circulation events, such as global warming. In a warmer climate, water vapor in the
atmosphere tends to increase, which destabilizes the atmosphere and enhances precipitation (Chou et al., 2012). Therefore,
most positive precipitation anomalies are expected from a theoretical point of view in spite of the associated atmospheric
circulation does not change too much.
*Acknowledgements*. This study was funded by the National Key Research and Development Program of China (Grant No.
2016YFC0401307) and the National Natural Science Foundation of China (Grant No. 41671026). We appreciate the editors and
anonymous reviewers for their constructive comments on improving the original manuscript.

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
