# Peer review of "Influence of three phases of El Niño-Southern Oscillation on daily precipitation regimes in China"

_Hydrology and Earth System Sciences, 2017_

## Short Comment (SC1) · 19 Feb 2018

I read with great interest this paper, but found it boring and meaningless. Actually, there are numerous researches addressing precipitation changes and relevant connections to ENSO, and other global climatic signals. However, I cannot find anything new or novel from this paper in terms of methods, idea and even results and conclusions. Besides, the authors of this paper presented statistical results only but no causes and physical mechanisms were discussed with enough evidences. Therefore, I cannot take this study as a real study. It is nothing but a simple statistical analyses. In general, this paper lacks novelty in methods, idea and even conclusions. No new findings can be found. What's more, it is a kind of repeated work, in this sense, I do think it is boring to read such a repeated work.

[Figure]

I cannot suggest acceptance because of its lowest quality and presentation quality.

---

## Referee Comment (RC1) · Anonymous Referee #1 · 12 Mar 2018

Review comments on Lv et al. "Influence o three phases of El Nino-Southern Oscillation on daily precipitation regimes in China".

Lv et al. analyzed the precipitation (and related indexes of extreme events, including drought) anomalies during various phases/types of ENSO events by using observational weather station data. The unique new findings from this study are the analyses on the extreme precipitation (and dry) events and the inclusion of the CP events for these analysis and comparisons among them. A big uncertain from this study is using observed daily precipitation data to accounting for the rainfall intensity and frequencies and using the annual period to account the total precipitation (or indexes), rather than the real duration of ENSO events/phases. Another limitation is the authors failed to explain the data/results with possible processes or mechanism; which is more meaningful

to enhance out current understanding on the teleconnections during ENSO events.

Minor comments: Line 86: 713 stations; line 91: 719 stations Table 1. 1. Add definition of "wet days"; 2. Need rephrase the definition of R95P (i.e. need to clarify the time step of precipitation and describe what is the 95th percentile of multi-year average).

Line 104: need more detailed descriptions on Mann-Whitney U test. Fig. 2: Line 110: define the "developing years" and "decaying years", and describe their physical meanings. In some certain years, they are both belonging to stages of either developing or decaying years of different phases (EP, CP, or LN). Will this affect the conclusion? Line 130: Should add "decaying" between "CP" and "phases"? Line 135: define "precipitation frequency" and define "precipitation intensity"; are they counted for each precipitation event or each individual rainy day? Line 159: add "(Fig. 4)" after "while more than 70% experienced negative anomalies in precipitation frequency"; Line 188: delete "but also increase": Line 192-193: Rephrase the sentence "Although a positive anomaly ...". Line 206: the conclusion that "ENSO events triggered larger changes in both the frequency and intensity of precipitation events and the occurrence of precipitation extremes than during non-ENSO periods" is not support by the data; to support this claim, a comparison with precipitation indexes from "non-ENSO" period should be conducted. Line 226: add "(Fig. 6a)" after "occurred during CP decaying years": and line 227: add "(Table 3)" after "observed negative anomaly in annual precipitation". Line 227-229: rephrase this sentence. The main point is not clear. Line 247: may need briefly describe the "anomalous Western North Pacific (WNP) anti-cyclone"; how it happens and what its effect. Line 258: 1. what is "these and other regions"? 2. Please explain what is "high incidence of anomalies". Line 257-259: need explain why "the continental climate zone (NW) is more sensitive to ENSO events than these and other regions".

---

## Referee Comment (RC2) · Anonymous Referee #2 · 12 May 2018

The paper attempts to find the impact (if any) of the El Niño-Southern Oscillation (ENSO) on daily rainfall in China. For doing so, it uses rainfall data from more than 700 stations (1960-2013) across the country. It is known that establishing a link between ENSO and rainfall at a specific region is not an easy task, since there are a lot other variables in play.

Major comments: The index used for the selection of ENSO years needs a careful thought. A common index to use to identify a given ENSO year is Multivariate ENSO Index (MEI, https://www.esrl.noaa.gov/psd/enso/past\_events.html). Using different indices for the identification may result in (wrongly) classifying the same year as an El Nino, la Nina or a neutral year. This seems to be the case in this paper where several of the years in Table 2 do not correspond with the events identified using MEI. Since the

entire analysis and interoperation of the result strongly depends on the ENSO event identification, the authors should make sure that they are using an accurate index. Furthermore, ENSO events can last longer than 1 calendar year often spanning Fall of one year to Summer of the following year. Some precipitation indices presented in the paper are wrongly termed as "new" (line 88), while they have been used by the WMO and several other studies (see for example Zhang et al., 2011, Alexander et al., 2013). The paper needs a wider literature review on hydrological impacts of ENSO. There are important works which are overlooked, leading to mistakenly labelling the central pacific El Nino as a new type (Line 67-70) while it has been recognised since, at least, 2005 (Larkin and Harrison, 2005; Hu et al., 2016). Also see Emerton et al. (2017) for the likelihood of ENSO-driven global flood hazard.

Other comments: - The three phases of ENSO as commonly known are Neutral, El Niño or La Niña. It is a bit confusing the way it is used in the title. - Line 47-50: break the sentence into two. - Line 84: remove one bracket - Line 87: what do you mean by amount? Possibly to mean "duration, intensity and frequency"? - Line 96: 2011 is not new. - Table 2. Number of wet days is not really extreme. - What is the threshold for the definition of we days (e.g., 0mm/day)? - Are the El Nino/La Nina years excluded from the calculation of the multi-year average? - The blue shading in figures 2, 4, and 5 are interchanged. The dark blue should correspond with the intense rainfall.

**References**

Emerton, R., Cloke, H. L., Stephens, E. M., Zsoter, E., Woolnough, S. J. and Pappenberger, F. (2017) Complex picture for likelihood of ENSO-driven flood hazard. Nature Communications, 8. 14796. ISSN 2041-1723 doi: https://doi.org/10.1038/ncomms14796

Larkin NK, Harrison DE (2005) On the definition of El Niño and associated seasonal average US weather anomalies. Geophys Res Lett 32:L13705. doi:10.1029/2005GL022738
Hu et al., 2016: Contrasting the eastern Pacific El Niño and the central Pacific El Niño: processŚbased feedback attribution, Clim Dyn (2016) 47:2413–2424, DOI 10.1007/s00382-015-2971-9

Zhang, X.; Alexander, L.; Hegerl, G.C.; Jones, P.; Tank, A.K.; Peterson, T.C.; Trewin, B.; Zwiers, F.W. Indices for monitoring changes in extremes based on daily temperature and precipitation data. Wiley Interdiscipl. Rev. Clim. Chang. 2011, 2, 851–870.

Alexander et al., 2013, ClimPACT, Indices and software, www.wmo.int/pages/prog/wcp/ccl/.../ETCRSCI\_software\_documentation\_v2a.doc

---

## Author Comment (AC1) · 22 Jun 2018

**Reply to the comment of Dr. Q. Zhang:**

We pay our highest tributes to his valuable comment. We would like to mention that we noticed previous studies on ENSO-climate teleconnections and reviewed the progress in their theory and frontier research issue in the introduction part of our manuscript. The following are the detailed explanations for our scientific focus, methodology, and main findings.

\*\*\*\*\*\*\*\*\*\*\*\*\*\*\*\*\*\*\*\*\*\*\*\*\*\*\*\*\*\*\*\*\*\*\*\*\*\*\*\*\*\*\*\*\*\*\*\*\*\*\*\*\*\*\*\*\*\*\*\*\*\*\*\*\*\*\*\*\*\*\*\*\*\*\*\*\*\*\*\*\*\*

*'I read with great interest this paper, but found it boring and meaningless. Actually, there are numerous researches addressing precipitation changes and relevant connections to ENSO, and other global climatic signals. However, I cannot find anything new or novel from this paper in terms of methods, idea and even results and conclusions.'*

As an influential ocean-atmosphere phenomenon, ENSO has been reported to exert enormous changes on climate and hydrology over the world, especially around the Pacific Rim (e.g. Gershunov and Barnett, 1998; Gong and Wang, 1999; Yeh et.al, 2009). In China, ENSO dominates parts of the abnormal signals in monsoon systems. Various studies have extensively documented the teleconnections between ENSO and precipitation anomalies in different spatio-temporal scales (e.g. Lin and Yu, 1993; Huang and Wu, 1989; Zhou and Wu, 2010). However, most studies have focused on changes in annual or seasonal precipitation amount related to ENSO rather than changes in individual precipitation events, although possible shifts in characteristics of precipitation events, e.g. frequency and intensity, have been highlighted in studies of climate change around China (Zhai et al., 2005). The anomalies in precipitation amount derive from anomalies in its frequency, intensity or the joint distribution, but precipitation events and their extremes are direct indicators more relevant to hydrology than annual and monthly amounts. What's more, precipitation extremes, such as consecutive wet days and dry spells, which directly relate to droughts and floods, have rarely been addressed in previous studies of ENSO over China.

Thus, we use a comprehensive set of precipitation indices in this study to reach our main objectives: (1) describe ENSO-induced precipitation anomalies between developing and decaying stages; (2) compare the anomalies of precipitation frequency and intensity with those of annual amount; and (3) propose possible changes in precipitation extremes responsible for these anomalies.

In the goal-oriented analysis, this study highlights the new type of El Niño events, Central Pacific El Niño, and differences between developing and decaying stages.

Our innovation is to integrate potential anomalies in characteristics of precipitation events with those in precipitation amount to inform climate and hydrology policy. We respect his opinion about our research innovation, but we think our study have proposed very interesting research gaps where a sufficient attention was paied in previous studies. We hope he was willing to reread our manuscript and rethink the scientific problems we put forward.

\*\*\*\*\*\*\*\*\*\*\*\*\*\*\*\*\*\*\*\*\*\*\*\*\*\*\*\*\*\*\*\*\*\*\*\*\*\*\*\*\*\*\*\*\*\*\*\*\*\*\*\*\*\*\*\*\*\*\*\*\*\*\*\*\*\*\*\*\*\*\*\*\*\*\*\*\*\*\*\*\*\*\*\*\*\*\*\*

*'Besides, the authors of this paper presented statistical results only but no causes and physical mechanisms were discussed with enough evidences. Therefore, I cannot take this study as a real study. It is nothing but a simple statistical analyses. In general, this paper lacks novelty in methods, idea and even conclusions. No new findings can be found. What's more, it is a kind of repeated work, in this sense, I do think it is boring to read such a repeated work.'*

In the methodology, we adopt a commonly used way to obtain the anomalies of signals and test the anomalies by the nonparametric Mann–Whitney U (Teegavarapu et al., 2013). Although there is very limited creativity in our methodology, it is worth mentioning that meteorological stations from National Meteorological Centre in China have undergone rigorous selection process into our dataset to omit possible non-climatic noises (Qian and Lin, 2005; Qu et.al.2016). This process is not an easy task, includes extreme value and consistency check, spatial outliers test, and homogeneity test, but can significantly improve the quality of long-term climatic data. We think these methods are sufficient to discover the ENSO-induced changes in our objectives.

Many interesting findings are discovered in the anomalies of precipitation events. For example, Eastern Pacific (EP) El Niño caused less precipitation in developing years and more precipitation in decaying years, but a clear pattern was only found in decaying Central Pacific (CP) El Niño (line 114-line 132). This can be further explained in detail (line 151-line 172). Anomalies of the amount in EP roughly paralleled anomalies in frequency and intensity, but the anomalies were altered in CP. In CP decaying years, negative anomalies in frequency (Southern China) and positive anomalies in intensity (Northern China) resulted in the total pattern, that means the anomalies of the amount in different regions were dominant by different contexts of precipitation events. In CP developing

years, however, clear anomalies in characteristics of precipitation events didn't induce obvious anomalies in the amount across China. Our study provides in-depth explanation in the ENSO-induced precipitation anomalies which is little mentioned in previous studies. More findings can be derived from such comparisons above.

Moreover, this study explains what precipitation extremes ENSO events have triggered and what linkages are there with the anomalies of the amount. For example, in EP and CP decaying years and in LN developing years, the number of very wet days (R95p), the maximum rainfall in one day (Rx1d), and the number of consecutive wet days (CWD) all increased, which supplemented the linkages between anomalies of the amount and those of frequency and intensity. In addition, risk of floodings and droughts may be directly revealed by anomalies of precipitation extremes, such as probable droughts in EP developing years while probable floodings in EP decaying years.

The regional distributions of the anomalies in the contexts of precipitation events are very complex, and thus the anomalies of precipitation amount should be treated with care to further inform climate and hydrology policy, such as adaptation efforts ahead of extreme events. Regionally, we found that the continental climate zone of China is more sensitive to ENSO than other regions based on the region's high incidence and magnitude of anomalies in precipitation events. Possible driving mechanisms behind the spatial differences may be related to the anomalous Western North Pacific anti-cyclone and variability of East Asian summer monsoon, which is further analysed in our discussion and conclusion. In summary, our study puts forward an outline to explain the anomalies of precipitation in detail and provides a means of climate prediction on a daily time scale. We think lots of further researches are needed to advance our understanding after this study.

\*\*\*\*\*\*\*\*\*\*\*\*\*\*\*\*\*\*\*\*\*\*\*\*\*\*\*\*\*\*\*\*\*\*\*\*\*\*\*\*\*\*\*\*\*\*\*\*\*\*\*\*\*\*\*\*\*\*\*\*\*\*\*\*\*\*\*\*\*\*\*\*\*\*\*\*\*\*\*\*\*\*\*

*'I cannot suggest acceptance because of its lowest quality and presentation quality.'*

Finally, we respect all his opinions, but we hope he was willing to reconsider our innovation in ideas and findings and accept our manuscript for this publication.

We express sincere thanks to him for his efforts again!

---

## Author Comment (AC2) · 23 Jun 2018

**Reply to the comments of Anonymous Referee #1.**

We thank Referee #1 for his/her valuable comment. Our point-to-point reply raised by the referee are provided below. Please note that the referee's comment will be presented in italics, preceded by a "C", while the corresponding authors' responses will be presented in normal typeface, preceded by an "R". Please note that the line numbers provided in this reply are from the original version of our manuscript. A revised version of the manuscript has been placed at the end of this document and is a result of revisions made following comment from Dr. Q. Zhang, and Referee #1.

**General comment**

*C1: Lv et al. analyzed the precipitation (and related indexes of extreme events, including drought) anomalies during various phases/types of ENSO events by using observational weather station data. The unique new findings from this study are the analyses on the extreme precipitation (and dry) events and the inclusion of the CP events for these analysis and comparisons among them. A big uncertain from this study is using observed daily precipitation data to accounting for the rainfall intensity and frequencies and using the annual period to account the total precipitation (or indexes), rather than the real duration of ENSO events/phases. Another limitation is the authors failed to explain the data/results with possible processes or mechanism; which is more meaningful to enhance out current understanding on the teleconnections during ENSO events.*

**R1:** We would like to thank the referee for this review. Our innovation is to integrate potential anomalies in characteristics of precipitation events with those in precipitation amount to inform climate and hydrology policy. We adopt the observed daily precipitation dataset (Version 3) compiled by the National Meteorological Center in China. It contained 819 stations for the period from 1 January 1951 to 31 December 2013. This dataset is well distributed across China, including the Tibetan plateau, and is widely used in many studies. Although there is a big uncertainty using daily-scale data to define precipitation events, this definition has been a commonly used way to detect climate change (Brunetti et al., 2004; Karl & Knight, 1998; Liu et al., 2005). It is worth mentioning that the meteorological stations have undergone rigorous selection process into our dataset to omit possible non-climatic noises (Qian and Lin, 2005; Qu et.al.2016). This process includes extreme value and consistency check, spatial outliers test, and homogeneity test, and thus can significantly improve the quality of long-term climatic data. We think these methods are sufficient to discover the ENSO-induced changes in our objectives. In addition, the developing of ENSO events are very complex and varies with strength. Annual scale is one of the commonly used periods to detect the ENSO events. Although annual scale seems fairly large, we discover many interesting findings by this scale. Monthly scales or even more detailed periods can be put forward, but we intend to draw an outline to explain the anomalies of precipitation in daily scale, and thus the annual scale is chosen in our study. Possible driving mechanisms behind the anomalies of precipitation events are discussed in our discussion and conclusion (Line 246 -Line 254 and Line 278-Line 283). We highlight the importance of anomalous Western North Pacific anti-cyclone and variability of East Asian summer monsoon in this study, which is probably the reason causing the regional patterns we find. This analysis is derived from plentiful findings in previous studies. The direct evidence is hard to provide as the daily weather system is very complex to analyse and the needed data are also hard to get. We think lots of further researches are needed to advance our understanding after our study.

**Specific comment**

*C2: Line 86: 713 stations; line 91: 719 stations Table 1. 1. Add definition of "wet days"; 2. Need rephrase the definition of R95P (i.e. need to clarify the time step of precipitation and describe what is the 95th percentile of multi-year average). Line 104: need more detailed descriptions on Mann-Whitney U test.*

**R2:** There are 713 stations in our dataset, sorry for the mistake in line 91. In Table 1.1, wet days means the day with precipitation $> 0$ (data with trace amount have been deleted); definition of $95^{th}$ percentile of multi-year daily average has been stated clearly in the note of the table: the 95 quantiles of the daily precipitation distribution over the multi-year, represented by 1971–2000 (percentiles near 100 represent very intense precipitation). More detailed descriptions on Mann-Whitney U test are added in line 107: The Mann–Whitney U test is a nonparametric test applied to site data that does not conform to normality even after several different transformations are carried out (Teegavarapu et al., 2013). It tests whether two series are independent samples from different continuous distributions. One series represents precipitation series under a type of ENSO event (EP, CP, or LN), and the other series represents precipitation series under normal years.

*C3: Fig. 2: Line 110: define the "developing years" and "decaying years", and describe their physical meanings. In some certain years, they are both belonging to stages of either developing or decaying years of different phases (EP, CP, or LN). Will this affect the conclusion? Line 130: Should add "decaying" between "CP" and "phases"?*

**R3:** The definition of developing years and decaying years is added: as EP El Niño evolves, the positive SST anomalies expands latitudinally and negative signals eastward, and then reaches it maximum amplitude in autumn and winter (Feng et al., 2011); This year is defined as the developing year in this study. Finally, the warm SST anomalies disappears and is replaced by cool anomalies in the eastern Pacific, which occurs during summer of the next year (the decaying year); We have added "decaying" between "CP" and "phases" in line 130. We don't think this kind of overlap in years (*both belonging to stages of either developing or decaying years of different phases*) affects the final conclusion. The overlap depends on the developing of El Niño, which may reflect the connections among EP, CP, and LN. For example, in the decaying stage of EP El Niño, cool SST anomalies in the eastern Pacific will replace the warm anomalies, which probably denotes a LN event.

*C4: Line 135: define "precipitation frequency" and define "precipitation intensity"; are they counted for each precipitation event or each individual rainy day? Line 159: add "(Fig. 4)" after "while more than 70% experienced negative anomalies in precipitation frequency"; Line 188: delete "but also increase"; Line 192-193: Rephrase the sentence "Although a positive anomaly ...".*

**R4:** The definition of "precipitation frequency" and "precipitation intensity" is added in Table 1: intensity is average precipitation on wet days (day with precipitation > 0, data with trace amount have been deleted) and frequency is annual count of wet days; The citation of fig.4 is added after the sentence "while more than 70% experienced negative anomalies in precipitation frequency"; The sentences in line 188 and line192-193 has been revised.

*C5: Line 206: the conclusion that "ENSO events triggered larger changes in both the frequency and intensity of precipitation events and the occurrence of precipitation extremes than during non-ENSO periods" is not support by the data; to support this claim, a comparison with precipitation indexes from "non-ENSO" period should be conducted.*

**R5:** In our methodology, the anomalies of precipitation indices are derived from the multi-year average. So, the

"non-ENSO periods" is an incorrect expression. We have rephrased this sentence. Thanks for his/her responsibility and carefulness.

*C6: Line 226: add "(Fig. 6a)" after "occurred during CP decaying years"; and line 227: add "(Table 3)" after "observed negative anomaly in annual precipitation". Line 227-229: rephrase this sentence. The main point is not clear.*

**R6:** The citations of fig.6a and Table 3 have been added after the sentence. The sentence has been rephrased to explain the strong regional sensitivity in NW of China.

*C7: Line 247: may need briefly describe the "anomalous Western North Pacific (WNP) anti-cyclone"; how it happens and what its effect.*

**R7:** More detailed description about the Western North Pacific (WNP) anti-cyclone has been added in our manuscript: ENSO has strong effects on the anti-cyclone activity (e.g. number, genesis location, track, landfall frequency, and intensity) in WNP. Variation of the anti-cyclone activity directly influences on water vapor transport over China. As a result, the frequency and intensity of precipitation events in China would change.

*C8: Line 258: 1. what is "these and other regions"? 2. Please explain what is "high incidence of anomalies". Line 257-259: need explain why "the continental climate zone (NW) is more sensitive to ENSO events than these and other regions".*

**R8:** We have rephrased the original sentence as " However, in this study, we found that the continental climate zone (NW) is more sensitive to ENSO events than the other regions over China based on the region's magnitude of anomalies in annual precipitation and precipitation events (Fig. 3, Fig.6, and Table 3). For example, the NW region experienced the largest anomalies in annual precipitation during all phases of ENSO events. The NW region also demonstrated high sensitivity to the new type of El Niño, CP El Niño." In our manuscript, we repeat the related finding in annual precipitation, and further compare it with those of precious studies (e.g. Ward et al., 2014; Hui et al., 2006). In fact, this conclusion can be derived from our analysis in results, where lots of evidence to proof strong sensitivity of NW zone can be found.

Finally, we express sincere thanks to him/her for the efforts again!

**Additional References**

Brunetti, M., Maugeri, M., Monti, F., & Nanni, T. (2004). Changes in daily precipitation frequency and distribution in Italy over the last 120 years. *Journal of Geophysical Research: Atmospheres, 109*(D5).

Karl, T. R., & Knight, R. W. (1998). Secular trends of precipitation amount, frequency, and intensity in the United States. *Bulletin of the American Meteorological society, 79*(2), 231-241.

Liu, B., Xu, M., Henderson, M., & Qi, Y. (2005). Observed trends of precipitation amount, frequency, and intensity in China, 1960–2000. *Journal of Geophysical Research: Atmospheres, 110*(D8).

[revised manuscript text omitted]
 means the 95 quantiles of the daily precipitation distribution over the multi-year, represented by 1971–2000 (percentiles near 100 represent very intense precipitation).

In this study, two new indices created by Ren and Jin by transforming the traditionally-used Niño3 and Niño4 indices (2011)

were used to distinguish between CP and EP El Niño phases. La Niña years were identified using the methods of McPhaden and Zhang (2009). The ENSO events (1960–2013) analyzed in this study are displayed in Table 2. As EP El Niño evolves, the positive SST anomalies expands latitudinally and negative signals eastward, and then reaches it maximum amplitude in autumn and winter (Feng et al., 2011). This year is defined as the developing year in this study. Finally, the warm SST

anomalies disappears and is replaced by cool anomalies in the eastern Pacific, which occurs during summer of the next year (decaying year). Precipitation indices were calculated in both ENSO developing and decaying years. Indices for precipitation anomalies were analysed as follows:

$$A_{ij} = \frac{\overline{PI_{ij}} - \overline{PA_{ij}}}{\overline{PA_{ij}}} ,$$ (1)

Where $\overline{PI_{ij}}$ is the average of the $i$ precipitation index in the $j$ meteorological station in a specific time period, and $\overline{PA_{ij}}$

is the average of the $i$ precipitation index in the $j$ station in the multi-year average (represented by 1971–2000).

**Table 2.** ENSO years from 1960 to 2013

| Phase | Eastern Pacific (EP) El Niño | Central Pacific (CP) El Niño | La Niña (LN) |
|---|---|---|---|
| Year | 1963 1965 1969 1972 1976 1982 1986 1991 1997 2006 | 1968 1977 1987 1994 2002 2004 2009 | 1964 1967 1970 1973 1975 1984 1988 1995 1998 2007 2010 |

The Mann–Whitney U test is a nonparametric test applied to site data that does not conform to normality even after several different transformations are carried out (Teegavarapu et al., 2013). It tests whether two series are independent samples from different continuous distributions. One series represents precipitation series under a type of ENSO event (
[revised manuscript text omitted]
 (Feng et al., 2011), as ENSO has strong effects on the anti-cyclone activity (e.g. number, genesis location, track, landfall frequency, and intensity) in WNP. Variation of the anti-cyclone activity directly influences on water vapor transport over China. As a result, the frequency and intensity of precipitation events in China would change. This study reveals that due to increased WNP anti-cyclones during EP decaying phases, annual precipitation and the frequency and intensity of precipitation events increased, especially in northern China. During CP developing years, the anomalous WNP anti-cyclone weakens and causes no significant rainfall anomalies in China. However, during summer of CP decaying years, the WNP anti-cyclone re-invigorates and extends north-westward toward inland regions (Wu et al., 2003). As a result, plentiful moisture is transported to northern and northwestern China. Also, precipitation extremes may change in response to the different duration and magnitude of precipitation events.

The climate of several regions in China are vulnerable to ENSO events via teleconnections, such as the South China Sea (Qu et al., 2004; Rong et al., 2007; Zhou and Chan, 2007; Liu et al., 2011) and Yangzi River (Huang and Wu, 1989; Tong et al., 2006; Zhang et al., 2007; Zhang et al., 2015). However, in this study, we found that the continental climate zone (NW) is more sensitive to ENSO events than the other regions over China based on the region's magnitude of anomalies in annual precipitation and precipitation events (Fig. 3, 
[revised manuscript text omitted]

157-167, 2007.

---

## Author Comment (AC3) · 24 Jun 2018

**Reply to the comments of Anonymous Referee #2.**

We thank Referee #2 for his/her valuable comment. Our point-to-point reply raised by the referee are provided below. Please note that the referee's comment will be presented in italics, preceded by a "C", while the corresponding authors' responses will be presented in normal typeface, preceded by an "R". Please note that the line numbers provided in this reply are from the original version of our manuscript. A revised version of the manuscript has been placed at the end of this document and is a result of revisions made following comment from Dr. Q. Zhang, Referee #1, and Referee #2.

*C1: The paper attempts to find the impact (if any) of the El Niño-Southern Oscillation (ENSO) on daily rainfall in China. For doing so, it uses rainfall data from more than 700 stations (1960-2013) across the country. It is known that establishing a link between ENSO and rainfall at a specific region is not an easy task, since there are a lot other variables in play.*

**R1:** We would like to thank the referee for this review. Although various studies have extensively documented the teleconnections between ENSO and precipitation anomalies in different spatio-temporal scales, we have proposed very interesting research gaps where a sufficient attention was paid in previous studies. In this study, we intend to integrate potential anomalies in characteristics of precipitation events with those in precipitation amount to inform climate and hydrology policy. We have made lots of analysis and attempted to reach our objectives in the end.

**Major comment**

*C2: Major comments: The index used for the selection of ENSO years needs a careful thought. A common index to use to identify a given ENSO year is Multivariate ENSO Index (MEI, https://www.esrl.noaa.gov/psd/enso/past_events.html). Using different indices for the identification may result in (wrongly) classifying the same year as an El Nino, la Nina or a neutral year. This seems to be the case in this paper where several of the years in Table 2 do not correspond with the events identified using MEI. Since the entire analysis and interoperation of the result strongly depends on the ENSO event identification, the authors should make sure that they are using an accurate index. Furthermore, ENSO events can last longer than 1*

*calendar year often spanning Fall of one year to Summer of the following year.*

**R2:** Exactly! The definition of ENSO events determines the accuracy of the results and ultimately the conclusion. We have noticed the different indices for the identification of ENSO events, so we put lots of efforts into it in the beginning of our study.

(1) LN events.

We adopt the method of the NOAA oceanic Niño index (ONI, http://origin.cpc.ncep.noaa.gov/products/analysis_monitoring/ensostuff/ONI_v5.php). It is one of the commonly used indices around the world, which is defined as a 3-month running mean of SST anomalies over 5 consecutive months in the Niño -3.4 region (5°N–5°S, 90–150°W). A detailed list can be found in Table 1 of McPhaden and Zhang (2009). We further add year 2010 according to the ONI table in http://origin.cpc.ncep.noaa.gov/products/analysis_monitoring/ensostuff/ONI_v5.php.

(2) CP and EP events.

Traditionally, El Niño event is classified as a CP type if SST anomalies averaged over the Niño 4 region are greater than those averaged over the Niño 3 region and vice versa for the EP type (Kug et al., 2009; Yeh et al., 2009). However, Ren and Jin (2011) considered that they cannot effectively separate the two types of El Niño, as these two indices are highly correlated in time. They proposed a new set of indices with little simultaneous correlation, by performing a simple transformation of the Niño 3 and 4 indices. Yu and Kim (2013) reviewed and compared the different indices used to identify the CP and EP events, including Niño 3 and Niño 4 indices, El Niño Modoki Index (EMI), and Ren and Jin (2011) (detailed information can be found in their article). The transformed indices of Ren and Jin (2011) are more sensitive to detect CP events in some cases, such 1987-1988 and 2002-2003. In this study, we adopt the method of Ren and Jin (2011) to identify the CP and EP events.

Thus, we think these methods are sufficient to discover the ENSO events for our study. Multivariate ENSO Index (MEI) is also an efficient method to identify the El Niño/ La Niña. However, it is not for the CP/EP events. The Central-Pacific (CP) type has sea surface temperature (SST) anomalies near the Date Line, and the Eastern-Pacific (EP) type has anomalies centered over the cold tongue. They are not equal to the neutral stages named by MEI. We have noticed the list of MEI events, but are confused by the years identified. For example, the period of 2010-2011 is a traditional La Niña, which emerged in year 2010 and gradually vanish in year 2011 (it is identified by other indices, and also presented by a movie in https://www.esrl.noaa.gov/psd/enso/past_events.html), but this year is not on the list.

*C3: Some precipitation indices presented in the paper are wrongly termed as "new" (line 88), while they have been used by the WMO and several other studies (see for example Zhang et al., 2011, Alexander et al., 2013).*

**R3:** These indices are not original found in our study. Sorry about the incorrect statement. We have rephrased the sentence.

*C4: The paper needs a wider literature review on hydrological impacts of ENSO. There are important works which are overlooked, leading to mistakenly labelling the central pacific El Nino as a new type (Line 67-70) while it has been recognised since, at least, 2005 (Larkin and Harrison, 2005; Hu et al., 2016). Also see Emerton et al. (2017) for the likelihood of ENSO-driven global flood hazard.*

**R4:** As is implied in our introduction, the Central-Pacific (CP) type develops in regions of warming SSTs in the Pacific near the International Date Line, while the Eastern-Pacific (EP) type has anomalies centered over the cold tongue. It is not a very newly finding about the emergency of CP events, but it has recently been emphasized as CP appears to induce climate anomalies that are distinctly different than those produced by the canonical EP. In addition, CP has been occurring more frequently in recent decades. Therefore, in our study, we emphasize it as the 'new type' flowing the other studies, such as McPhaden et al. (2006), Yeh et al. (2009), Ren and Jin, (2011), and Yu and Kim (2013).

Thanks for the introduction of the newly excellent article, Emerton et al. (2017). We have reviewed many studies about the precipitation extremes and hydrological consequences of ENSO. For example, Ward et al. (2014) examined peak daily discharge in river basins across the world to identify flood-vulnerable areas sensitive to ENSO. Perez et al. (2011) modelled non-contiguous and contiguous drought areas to analyze spatio-temporal drought development. Water storage is another index typically used to detect frequency and magnitude of droughts during ENSO events (Veldkamp et al., 2015; Zhang et al., 2015). The precipitation extremes can indicate the risk of floodings and droughts. As one of our objectives in this study, we intend to propose possible changes in precipitation extremes induced by ENSO responsible for the anomalies in precipitation amount, frequency, and intensity. However, as there is a nonlinearity between precipitation and flood magnitude, probabilities have large uncertainties due to accuracy of the data and clear differences between the hydrological analysis and precipitation (Emerton et al., 2017).

**Other comment**

*C5: Other comments: - The three phases of ENSO as commonly known are Neutral, ElNiño or La Niña. It is a bit confusing the way it is used in the title. -*

**R5:** We would like to emphasize the three types of ENSO (SP, EP, and LN) in the title. If you think the word 'phases' is very confused, we can consider to change it into 'types'.

*C6: Line 47-50: break the sentence into two. - Line 84: remove one bracket - Line 87: what do you mean by amount? Possibly to mean "duration, intensity and frequency"? - Line 96: 2011 is not new. - Table 2. Number of wet days is not really extreme. - What is the threshold for the definition of we days (e.g., 0mm/day)? .*

**R6: -**The sentences in line 47-50 and line 84 have been revised. -Precipitation amount means annual amount of precipitation. -We have rephrased it in new version. -The sentences in line 96 has been rephrased. -Number of wet days is the precipitation frequency, one of our indices in Table 1. -The threshold of the definition of wet days is 0 mm/day, as the trace precipitation has been deleted in the data processes. The description has been added in the Table 1.

*C7: Are the El Nino/La Nina years excluded from the calculation of the multi-year average?*

**R7:** The CP, EP and LN are included in our multi-year average (1971-2000). The 30-year time period can include the ENSO variable and other climatic factors, but we consider the long-time period can be used to represent the basic condition of local climate.

*C8: - The blue shading in figures 2, 4, and 5 are interchanged. The dark blue should correspond with the intense rainfall.*

**R8:** Thanks for his/her chariness and responsibility. We have updated the figure 2, 4, and 5.

Finally, we express sincere thanks to him/her for the efforts again!

**Additional References**

Emerton, R., Cloke, H., Stephens, E., Zsoter, E., Woolnough, S., & Pappenberger, F. (2017). Complex picture for likelihood of ENSO-driven flood hazard. *Nature communications, 8*, 14796.

Kug, J.-S., Jin, F.-F., & An, S.-I. (2009). Two types of El Niño events: cold tongue El Niño and warm pool El Niño. *Journal of Climate, 22*(6), 1499-1515.

[revised manuscript text omitted]
 means the 95 quantiles of the daily precipitation distribution over the multi-year, represented by 1971–2000 (percentiles near 100 represent very intense precipitation).

In this study, two indices created by Ren and Jin (2011) by transforming the traditionally-used Niño3 and Niño4 indices were used to distinguish between CP and EP El Niño phases. La Niña years were identified using the methods of McPhaden and Zhang (2009). The ENSO events (1960–2013) analysed in this study are displayed in Table 2. As EP El Niño evolves, the positive SST anomalies expands latitudinally and negative signals eastward, and then reaches it maximum amplitude in autumn and winter (Feng et al., 2011). This year is defined as the developing year in this study. Finally, the warm SST anomalies disappears and is replaced by cool anomalies in the eastern Pacific, which occurs during summer of the next year (decaying year). Precipitation indices were calculated in both ENSO developing and decaying years. Indices for precipitation anomalies were analysed as follows:

$$A_{ij} = \frac{\overline{PI_{ij}} - \overline{PA_{ij}}}{\overline{PA_{ij}}},$$

(1)

Where $\overline{PI_{ij}}$ is the average of the $i$ precipitation index in the $j$ meteorological station in a specific time period, and $\overline{PA_{ij}}$ is the average of the $i$ precipitation index in the $j$ station in the multi-year average (represented by 1971–2000).

**Table 2.** ENSO years from 1960 to 2013

| Phase | Eastern Pacific (EP) El Niño | Central Pacific (CP) El Niño | La Niña (LN) |
|---|---|---|---|
| Year | 1963 1965 1969 1972 1976 1982 1986 1991 1997 2006 | 1968 1977 1987 1994 2002 2004 2009 | 1964 1967 1970 1973 1975 1984 1988 1995 1998 2007 2010 |

The Mann–Whitney U test is a nonparametric test applied to site data that does not conform to normality even after several different transformations are carried out (Teegavarapu et al., 2013). It tests whether two series are independent samples from different continuous distributions. One series represents precipitation series under a type of ENSO event (
[revised manuscript text omitted]
 (Feng et al., 2011), as ENSO has strong effects on the anti-cyclone activity (e.g. number, genesis location, track, landfall frequency, and intensity) in WNP. Variation of the anti-cyclone activity directly influences on water vapor transport over China. As a result, the frequency and intensity of precipitation events in China would change. This study reveals that due to increased WNP anti-cyclones during EP decaying phases, annual precipitation and the frequency and intensity of precipitation events increased, especially in northern China. During CP developing years, the anomalous WNP anti-cyclone weakens and causes no significant rainfall anomalies in China. However, during summer of CP decaying years, the WNP anti-cyclone re-invigorates and extends north-westward toward inland regions (Wu et al., 2003). As a result, plentiful moisture is transported to northern and northwestern China. Also, precipitation extremes may change in response to the different duration and magnitude of precipitation events.

The climate of several regions in China are vulnerable to ENSO events via teleconnections, such as the South China Sea (Qu et al., 2004; Rong et al., 2007; Zhou and Chan, 2007; Liu et al., 2011) and Yangzi River (Huang and Wu, 1989; Tong et al., 2006; Zhang et al., 2007; Zhang et al., 2015). However, in this study, we found that the continental climate zone (NW) is more sensitive to ENSO events than the other regions over China based on the region's magnitude of anomalies in annual precipitation and precipitation events (Fig. 3, 
[revised manuscript text omitted]

---

## Author Response (AR2)

**Reply to the comments of Anonymous Referee #1.**

We thank Referee #1 for his/her valuable comment. Our point-to-point reply raised by the referee are provided below. Please note that the referee's comment will be presented in italics, preceded by a "C", while the corresponding authors' responses will be presented in normal typeface, preceded by an "R". Please note that the line numbers provided in this reply are from the original version of our manuscript and the references used in this reply can be found in the original version of our manuscript. A revised version of the manuscript has been placed at the end of this document and is a result of revisions made following comment Referee #1.

**General comment**

*C1: This revision answers most of my concerns and questions but I still have some concerns on the choice of period duration for these three phases of ENSO events, i.e. the developing and decaying years, which may really only cover couple of months, rather than entire year. Another concern is on the use of daily precipitation to represent the intensity of precipitation events. Daily time intensity surely is better than monthly rainfall but still has uncertainties to represent true variations for each individual event. The paper should add some discussions on this issue. The followings has some minor comments.*

**R1:** We would like to thank the referee for this insightful comment.

The peak period of ENSO events is about several months, but typically, their evolution can last longer than one entire year. For example, EP El Niño events tend to span two calendar years, evolving in boreal spring and reaching their peak magnitude in winter of the same year, before decaying into the following spring/summer (Feng et.al., 2011). The emerging and vanishing stages of CP El Niño and La Niña events also show period durations longer than one entire year (McPhaden and Zhang, 2009), and typically, the magnitude and direction of ENSO evolution changes from the emerging year to next year. There is a significant time lag between the responses of climate in China and ENSO evolution (Lu et.al., 2011; Wu et.al., 2004), probably due to the lagged responses of western North Pacific subtropical high or winter/summer monsoon. So, our study assumes the evolution of ENSO events will induce different responses of climate in China to different stages, especially developing and decaying stages. The different impacts of different ENSO evolution stages on the climate of China have also been discussed extensively in previous studies, such as Southern China winter-spring precipitation (Chen et.al., 2013), China rainfall in the decaying phases (Feng et.al., 2011), summer precipitation in China (Gao et.al., 2006), East Asian winter monsoon to the following summer monsoon (Chen et. al., 2013), winter rainfall in China (Zhou et.al. 2010), rainfall anomalies in East Asia during developing and decaying year (Wu et.al., 2004), etc. Our study is based on these articles with discussion on ENSO evolution and aims to discuss the different responses of climate in China to different stages of ENSO events.

Daily precipitation has been widely used to represent the intensity of precipitation events at regional scales or global scales (Zhai et.al., 2005; Zhang et.al., 2011; Xu et.al, 2011; Zhang and Cong, 2014). Daily precipitation has also been used to identify droughts, foods, or climate extremes among abundant studies, such as Fowler et.al. (1995), Zhai et.al. (2005), Zhang et.al. (2011), Ward et.al. (2014), and Chou wt.al (2012) (all references in our manuscript). We think that although daily precipitation has 'uncertainties to represent true variations for each individual event', analysing daily climatic data is still very meaningful to detect the ENSO-climate teleconnection. As more daily data becomes available through data producer efforts, similar analyses for other areas of the world, other stages of ENSO evolution, or other indexes of climate extreme will provide considerable information to better understand how the daily climate responds to the ENSO events.

**Minor comments**

*C2: Suggest replace China with "Mainland China" since no information covers Taiwan.*

**R2:** In the discussion section, summer monsoons over East Asia and TC activity over the WNP were analyzed in area including Taiwan. In fig.1, Taiwan belongs to the tropical and subtropical monsoon zone (South, S). In this study, the information of summer monsoons and TC activity at Taiwan were analyzed with the daily precipitation stations near this region to detect the ENSO-climate teleconnection of this region.

*C3: Line 18-19: Rephrase "ENSO events...". Does it mean that the precipitation during the period of ENSO events controls the magnitude of annual precipitation?*

**R3:**   Yes, this sentence means anomalies in annual precipitation were controlled by the changes in frequency and intensity of daily rainfall based on our analysis. We have rephrased the sentence as 'Further analysis revealed anomalies in frequency and intensity of rainfall accounted for anomalies in annual precipitation'.

*C4: Line 21: "ENSO events triggered MORE extreme...", need add "than ...";*

**R4:** We have rephrased the sentence as 'ENSO events tended to trigger extreme precipitation events.'

*C5: Line 31: Replace "vulnerable" to another word; Line 32: Is it real that "global annual rainfall drops during El Nino phases"? or should it be global annual rainfall over the land?*

**R5:** The word has been replaced by 'susceptible'. The words 'over the land' have been added.

*C6: Line 38: add "strong" between "anomalous" and "southwesterly winds"; Line 39: replace "insufficient" with "less";*

**R6:** The words have been added in the sentence.

*C7: Line 41-43: rephrase "As ENSO...". I don't understand what it means.*

**R7:** The sentence has been rephrased, and another sentence is added follow the previous statement.

*C8: Line 50: replace "annual rainfall days" with "wet days per year"; Line 50-53: These examples have nothing related with ENSO events (literally) and they may cause confusing.*

**R8:** The statement has been replaced. Sorry for the misquoted reference of Perez et.al. (2011). The related statement has been removed.

*C9: Line 67: better replace "may now exist" with "should be defined (or separated from canonical El Nino)". Line 68: add "discovered" between "new" and "El Nino";*

**R9:** The statement has been replaced and added, respectively.

*C10: Line 74: rephrase "The current study...";*

**R10:** The sentence has been changed into a more appropriate sentence.

*C11: Line 86: delete "data from"; replace "between" with "of"; Line 87: what is the difference between "amount" and intensity?*

**R11:** These sentences have been updated.

*C12: Table 1: either use "rainy" or "wet" day, not both. Rephrase the definition of "R95p" as "Total precipitation over wet days with larger than 95th percentile during the period of 1971-2000". Add more details on how DS and CWD is counted. The units of these two indices might be "count"?*

**R12:** Table 1 has been updated.

*C13: Lines 100-104 defined the developing and decaying year of EP El Nino phase, how about CP and La Nina*

*phases? It is still confusing how these "developing" and "decaying" years are identified and how much uncertainties comes from this (in particular its total length and overlapping period with other phases).*

**R13:** The definition of CP and La Nina phases has been added. As explained above, our study assumes the evolution of ENSO events will induce different responses of climate in China to different stages of ENSO events, especially developing and decaying stages. The emerging and vanishing stages of ENSO events tend to show period durations longer than one entire year, and typically, the magnitude and direction of ENSO evolution changes from the emerging year to next year. In addition, we don't think the overlap in years, such as a year both belonging to stages of either developing or decaying years of different phases, affects the final conclusion. The overlap depends on the developing of El Niño, which may reflect the connections among EP, CP, and LN.

*C14: Line 110; Rephrase the sentence for describing the Mann-Whitney U test.*

**R14:** A sentence has been added before the previous statement 'The significance of ENSO-induced precipitation anomalies is tested using a Mann–Whitney U approach.'

*C15: Line 122: replace ". At 80 of these stations the anomalies" with "and 80 of them" (need double check if these stations with significant signals all have negative signs or not);*

**R15:** The sentences have been rephrased.

*C16: Line 136: Explain what is the "result" in the expression of "as a result"?*

**R16:** The sentences have been rephrased as 'These results suggested'.

*C17: Line 138-140: Rephrase this sentence. Which is "CP phases"? Developing or decaying phases?*

**R17:** The sentence has been rephrased

*C18: Line 169: rephrase "a clear distribution pattern";*

**R18:** The related map of this sentence has been cited in the end of the sentence.

*C19: Line 143: replace "the whole of China" with "Mainland China as a whole";*

**R19:** The analysis is on the whole of China not just Mainland China .

*C20: Line 200: replace "fewer" with "smaller";*

**R20:** The sentence has been rephrased.

*C21: Line 216: This sentence hints that this study was conducted on "individual precipitation events" but it is not true. A clarification such as "daily precipitation events" might be more suitable.*

**R21:** The sentence has been rephrased as the 'daily precipitation events'.

Finally, we express sincere thanks to him/her for the efforts again!

**References**

References used in this reply can be found in our manuscript.

[revised manuscript text omitted]